

# *Vitis* flower types: from the wild to crop plants

João L. Coito[1], Helena G. Silva[2], Miguel J.N. Ramos[1], Jorge Cunha[3], José Eiras-Dias[3], Sara Amâncio[1], Maria M.R. Costa[2] and Margarida Rocheta[1]

[1] Linking Landscape, Environment, Agriculture and Food (LEAF), Instituto Superior de Agronomia, Universidade de Lisboa, Lisboa, Portugal
[2] Biosystems and Integrative Sciences Institute (BioISI), Plant Functional Biology Centre, University of Minho - Campus de Gualtar, Braga, Portugal
[3] Instituto Nacional de Investigação Agrária e Veterinária, Quinta d'Almoinha, Dois Portos, Portugal

Corresponding author
Margarida Rocheta,
rocheta@isa.ulisboa.pt

## ABSTRACT

*Vitis vinifera* can be divided into two subspecies, *V. vinifera* subsp. *vinifera*, one of the most important agricultural crops in the world, and its wild ancestor, *V. vinifera* subsp. *sylvestris*. Three flower types can be observed: hermaphrodite and female (on some varieties) in *vinifera*, and male or female flowers in *sylvestris*. It is assumed that the different flower types in the wild ancestor arose through specific floral patterns of organ abortion. A considerable amount of data about the diversity of sexual systems in grapevines has been collected over the past century. Several grapevine breeding studies led to the hypothesis that dioecy in *vinifera* is derived from a hermaphrodite ancestor and could be controlled by either, one or two linked genetic determinants following Mendelian inherence. More recently, experiments using molecular approaches suggested that these *loci* were located in a specific region of the chromosome 2 of *vinifera*. Based on the works published so far, its seems evident that a putative sex *locus* is present in chromosome 2. However, it is still not fully elucidated whether flower types are regulated by two linked *loci* or by one *locus* with three alleles. Nevertheless, several genes could contribute to sex determination in grapevine. This review presents the results from early studies, combined with the recent molecular approaches, which may contribute to the design of new experiments towards a better understanding of the sex inheritance in grapevine.

## INTRODUCTION

The cultivated grapevine (*Vitis vinifera* subsp. *vinifera* L., hereafter *vinifera*) belongs to Vitaceae. Among this family, species from the genus *Vitis* are found mainly in temperate zones of the Northern Hemisphere distributed between North America and eastern Asia. *V. vinifera* is the only *Vitis* species proved to have its origin in Europe. Fossil vestiges of *Vitis* grape seeds from European tertiary period sediments suggest that Vitaceae is around 55 million years old (*Tiffney & Barghoorn, 1976*). The survival of *Vitis* genus was accompanied with a rise in *Vitis* species and their distribution is consistent with the breaking

up of *Vitis* large populations by ice fronts, only surviving in refuges, the ideal conditions for speciation (*Adam-Blondon, Martinez-Zapater & Kole, 2016*). In the case of *V. vinifera* the refuges corresponded to the regions stretching from the western Himalayas to the Caucasus, where the origin of the current *vinifera* is hypothesised to have occurred.

Evolutionary genomic analysis showed that *vinifera* diverged from *V. vinifera* subsp. *sylvestris* (hereafter *sylvestris*) around 22,000 to 30,000 B.C. (*Zhou et al., 2017*). *V. vinifera* subsp. *sylvestris* is considered the wild grapevine from which *vinifera* was domesticated (*Zohary, Hopf & Weiss, 2000*) in the region between the Black Sea and Iran, around 4,000 years B.C. (*McGovern, 2004*). However, more recent archaeological samples from the south Caucasus, suggest that wine making go as far as 8,000 years (*McGovern et al., 2017*). Therefore, the domestication process might have occurred earlier than the records show. The domestication and cultivation of *vinifera* then spread to what is today the Palestine, southern Lebanon and Jordan (*McGovern, 2004*). *V. vinifera* subsp. *vinifera* (hereafter *vinifera*) continued its propagation and in 3,000 B.C. it appeared in Asia Minor, Southern Greece, Crete and Cyprus. By 2,000 B.C. *vinifera* was found in areas surrounding the Mediterranean Sea, including southern Italy and the Balkans. From there, it expanded to the whole Mediterranean region including France, Spain and Portugal (*Arroyo-Garcia et al., 2006*; *McGovern, 2004*). The domestication of *vinifera* is linked with the discovery of wine production, as well as, other products such as raisins, table grapes and grape vinegar, making this species a major agronomic and economic crop worldwide (*Hardie, 2000*; *Royer, 1888*).

The last classification of the Vitaceae includes 14 genera (*Soejima & Wen, 2006*). *Vitis* genus itself has two subgenera: *Vitis* and *Muscadinia* (*Vitis rotundifolia*), both producing edible fruits (*Olien, 1990*). The distinction through karyological analysis showed $2n = 2x = 40$ for muscadines (subgenus *Muscadinia*) and $2n = 2x = 38$ for bunch grapes (subgenus *Vitis*). Despite the karyotype difference, crossing between *V. vinifera* and *V. rotundifolia* is viable. However there is a high sterility in the F1 hybrids with few or no viable seeds. This might be explained by the phylogenetic proximity between the two subgenera confirmed by chloroplastic gene markers (*Ingrouille et al., 2002*; *Soejima & Wen, 2006*).

The high number of *Vitis* chromosomes ($2n = 2x = 38$ and $2n = 2x = 40$) suggests a polyploid origin of the genomes, as generally assumed for other plant species (*Lewis, 1979*). Cytological analysis performed during pollen meiosis of F1 offspring of *V. vinifera* × *V. rotundifolia* led to the hypothesis of an allopolyploid origin of grapevine with three basic genomes. Two of those genomes would be common to *Vitis* and *Muscadinia* (with haploid contents of $n = 6$ and $n = 7$, respectively) and a third genome with $n = 6$ for *Vitis* and $n = 7$ for *Muscadinia* (*Patel & Olmo, 1955*). However, not only in situ hybridisation on *V. vinifera* chromosomes revealed a single ribosomal *locus* (*Haas & Alleweldt, 2000*), but the genetic maps did not give support to this hypothesis above (*Doligez et al., 2006*; *Lowe & Walker, 2006*). However, more recently the hypothesis of a polyploid origin of *Vitis* found strong support with the publication of the grapevine genome (*Jaillon et al., 2007*), describing the genome of *V. vinifera* as derived from a hexaploid ancestor with a presumable haploid number of chromosomes $n = 7$.

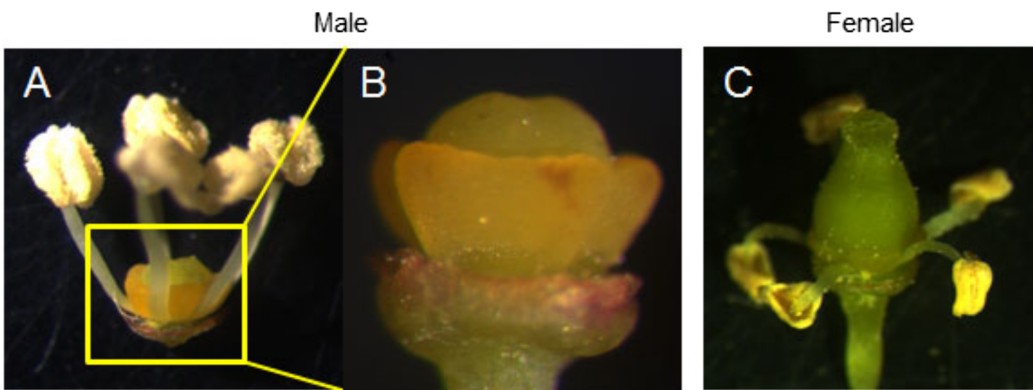

**Figure 1 The most striking morphological aspect of both flower types.** Dissected flowers from *Vitis vinifera sylvestris*. (A and B) Male flower of *V. v. sylvestris* without petals. (B) In male flowers, it is possible to observe the undeveloped pistil and in (C) females the reflex stamens, that produce infertile pollen. Photo credit: Margarida Rocheta.

## SURVEY METHODOLOGY

The authors of this review article are familiar with grapevine genetics, flower development, physiology and new generation sequencing techniques, having published several scientific manuscripts regarding the subject. Scientific articles cited on this review were searched on PubMed database, and Google scholar using: "grapevine" "*V. vinifera*", "flower development", "homeotic genes", "flower sex", "plant evolution", "domestication", "grapevine breeding" as keywords. Scientific articles, diverse in date of publishing, considered relevant on the subject were selected.

## THE SEXUAL SYSTEMS IN GRAPEVINE

Flowers of wild grapevines are mainly of two types: male (with suppressed pistils) (Fig. 1A–1B) or female (with reflexed stamens and infertile pollen) (Fig. 1C). However, in rare instances, female flowers producing fertile pollen appear in wild plants (*Valleau, 1916*).

Variations in the development and functional ability of stamens and pistils of the three grapevine flower types have been reported (*Stout, 1921*) although, these differences were considered as being quantitative rather than qualitative. Clear differences were observed among hermaphrodite flowers regarding filament length, anther size, pistil shape (Figs. 2A–2C) and seed number (*Stout, 1921*). In female flowers, *Stout (1921)* described a gradual phenotype regarding the reflexed stamen development (Figs. 2D–2H), with several grades of anther development, pollen grain features and dehiscence (Figs. 2D–2H). Pollen grains were irregular, shrivelled and an only few were able to germinate (*Stout, 1921*).

In accordance with a previous study (*Dorsey, 1912*), *Stout (1921)* showed that in male plants, the functional stamens may also vary in size (Figs. 2I and 2J) emphasising that these flowers exhibit different types of rudimentary pistils with or without stigmatic surfaces (Figs. 3A–3D). Carpels and rudimentary ovules were produced with seed coats not fully developed (*Dorsey, 1912*) and none of them were able to produce fruits (*Stout, 1921*).

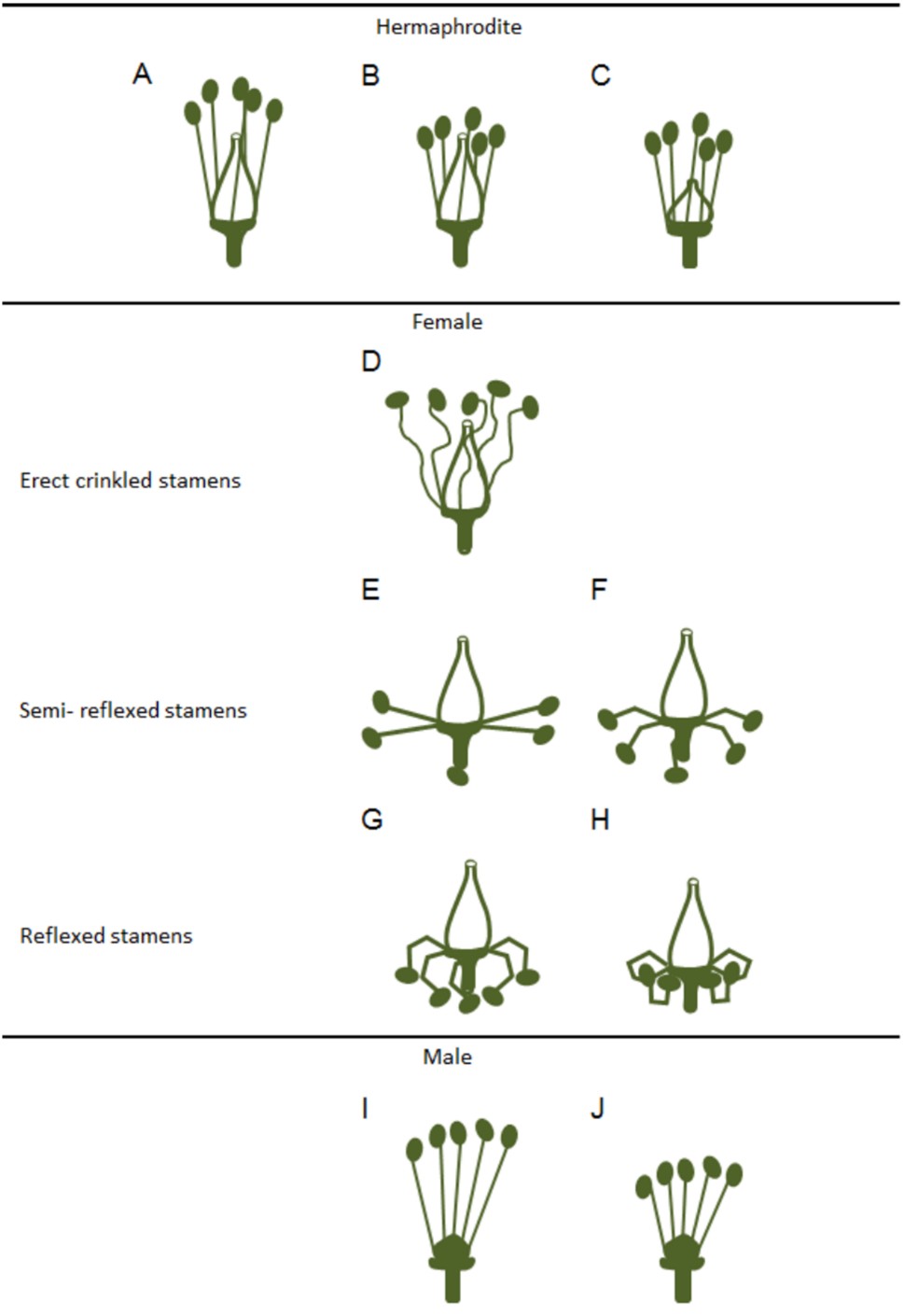

**Figure 2 Variations in stamens and pistils in *Vitis* flower types.** Scheme of some possible morphologic variations in stamens and pistils on hermaphrodite, female and male flowers (according to *Stout (1921)*). (A–C) Hermaphrodite plants hold both functional organs but might show variation in (B) stamens and (C) pistil size. (D–H) Female flowers show a functional pistil but several types of non-functioning stamens (producing infertile pollen). From (D) erect but crinkled stamens, (E and F) to semi-reflexed stamens, (G and H) and reflexed stamens. (I and J) Male plants exhibit several variations of non-functional pistils but fully functional stamens with variable size.

**Figure 3 Differences in the pistil of male plants.** Diagram of the morphologic differences in the pistil of male flowers (according to *Stout (1921)*). (A) Include well-formed but non-functional pistils, (B and C) pistils with different sizes, and (D) a non-functional pistil with the complete absence (X) of style and stigma.

In the middle of 20th century, *Levadoux (1956)* shed light on the differences between *vinifera* and *sylvestris* of western Eurasia and concluded that no morphological character allowed a specific differentiation between them since hermaphrodite plants could also occur in *sylvestris*. Later, several authors contributed to clarify the floral and reproductive development of *vinifera* and *sylvestris* in more detail (*Caporali et al., 2003*; *Gallardo et al., 2009*; *Popescu, Dejeu & Ocete, 2013*).

The morphological differentiation between male and female flowers of the dioecious grapevine can only be identified at a late stage of flower development, since at early stages a hermaphrodite development pattern is observed (*Caporali et al., 2003*). Although male flower pistils are not fully developed, the ovule development occurs, followed by megasporogenesis and megagametophyte formation (*Caporali et al., 2003*). The suppression of femaleness appears to be the consequence of the death of a specific layer, or layers, of nucellar cells and ovular integuments, which may represent the targets of a programmed cell death or apoptosis process (*Caporali et al., 2003*). In female flowers, the suppression of maleness appears to be the consequence of pollen sterility (*Caporali et al., 2003*). Comparing to male flowers microspores, female microspores have been shown to have an abnormal, round shape, without distinct colpi, and lack the typical thickenings (*Caporali et al., 2003*). The anthers of female flowers produce much less pollen grains (10–100) than the male (1,500–3,000) (*Gallardo et al., 2009*) and their size (polar axis × equatorial axis) is smaller in female flowers than in male ones (*Gallardo et al., 2009*; *Popescu, Dejeu & Ocete, 2013*).

## FLOWER TYPES IN GRAPEVINE: MENDELIAN APPROACH

Grapevine flower sex segregation was addressed using crosses between different *vinifera* varieties in the outset of the 20th century (*Hedrick & Anthony, 1915*). The authors

**Table 1 Sex determinants of hermaphrodites, males and females plants of grapevine according to several authors.**

|  | Valleau (1916) | Oberle (1938) | Avramov et al. (1967) | Negi & Olmo (1971) | Antcliff (1980) |
|---|---|---|---|---|---|
| Hermaphrodite | HF | so Sp/so sp | $S^h S^h$ | – | H |
|  | HH | so Sp/so Sp | $S^h S^f$ |  |  |
| Male | MH | So Sp/so Sp | – | $Su^F/Su^F$ | M |
|  | MF | So Sp/so sp |  | $Su^F/Su^m$ |  |
| Female | FF | so sp/so sp | $S^f$ | $Su^m/Su^m$ | F |

self-pollinated grapevines exhibiting upright stamens, reflexed stamens and performed crosses between these two types of grapevines. The interpretation of these results proposed that grapevine sex specification should be regulated by two determinants, F (female) and M (male) (*Valleau, 1916*) and assumed that female plants are homozygous, FF, and males are heterozygous, FM. In the hermaphrodites both determinants would be linked giving rise to heterozygous individuals represented by FH (of Hermaphrodite) (Table 1). The flower phenotypes from crosses between female and the different hermaphrodite types, allowed the conclusion that the hermaphrodites could be either FH or HH (*Valleau, 1916*) (Table 1). Several crosses between hermaphrodite and male plants revealed that MH male plants should also exist (*Valleau, 1916*). The most comprehensive work on the inheritance of flower sex in *vinifera* was carried out in 1938, and it was suggested that flower sex determination in grapes was due to a digenic linked inheritance (*Oberle, 1938*). The author assumed a homozygous condition in female plants, a heterozygous condition for the functionally male plants, and a heterozygous or a homozygous condition in the hermaphrodites. Also, it was proposed that the action of recessive genes resulted in male sterility and the action of dominant ones resulted in female sterility. *Oberle (1938)* named the dominant allele responsible for perfect pollen development as *Sp*, the recessive allele that inhibits pollen development as *sp*, the dominant allele that inhibits ovule development as *So* and the recessive allele responsible for perfect ovule development as *so* (Fig. 4). The genetic formulation for the different flower types would be then *so sp/so sp* for female, *So Sp/so sp* or *So Sp/so Sp* for male and *so Sp/so sp* or *so Sp/so Sp* for hermaphrodite plants (*Oberle, 1938*) (Table 1).

The flower phenotypes observed in crosses between male vs female, female vs hermaphrodite and male vs hermaphrodite plants followed a Mendelian inheritance, and the ratios calculated from the crosses performed by *Oberle (1938)* were similar to those obtained by *Valleau (1916)*. According to the authors, the crosses between hermaphrodites could only originate hermaphrodites or hermaphrodites and females in a 3:1 ratio, depending on the hermaphrodite genotype. Homozygous hermaphrodites crossed with males of any genotype originated 1:1 ratio of male to hermaphrodite progeny. Heterozygous hermaphrodites crossed to heterozygous males originated 2:1:1 ratio of male: hermaphrodite:female progeny. Moreover, the cross between heterozygous hermaphrodites (*so sp/so Sp*) and homozygous *Sp* male (*So Sp/so Sp*) plants only originated male and hermaphrodites in a 1:1 ratio. Female crossed with homozygous hermaphrodites the

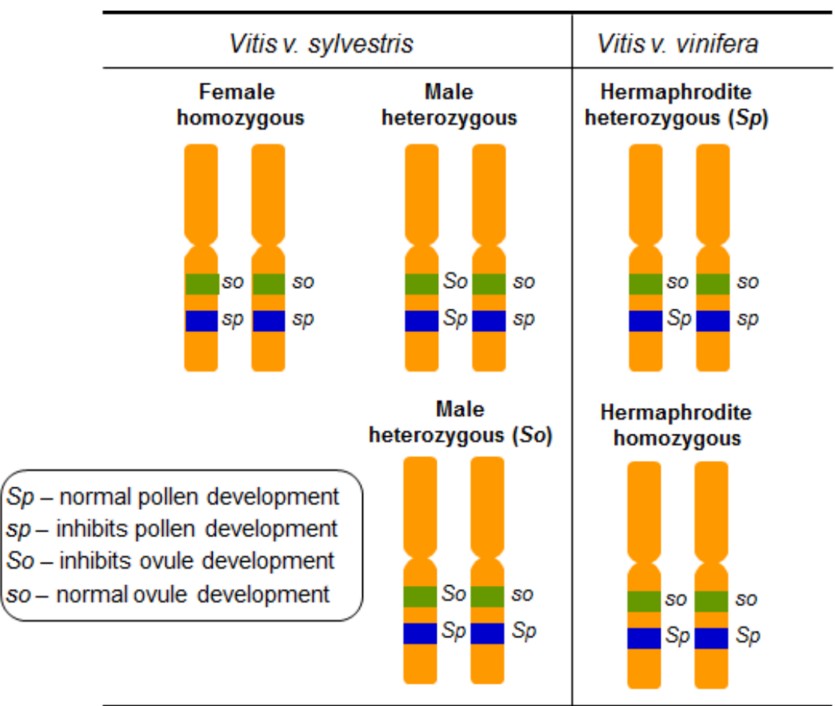

**Figure 4 Diagram illustrating the digenic determinants of sex inheritance in *Vitis*.** Illustration of the digenic determinants for flower sex in *Vitis* according to *Oberle (1938)*, using the *So/so* and *Sp/sp* genes and assuming a homozygous form for female plants (*so sp/so sp*) and heterozygous for male (*So Sp/so sp* and *So Sp/so Sp*) and hermaphrodite (*so Sp/so sp* and *so Sp/so Sp*) plants.

resulting progeny was 100% hermaphrodite. However, females crossed with heterozygous hermaphrodites the resulting proportion was 1:1 of female and hermaphrodites. Finally, the crosses between male and female plants, originated 1:1 proportion of males and hermaphrodites or males and females, depending on the possible male genotypes.

In a later work, several self-pollination and crosses with different cultivated hermaphrodite varieties were performed (*Avramov et al., 1967*). The authors obtained two sets of results: hermaphrodite and female plants in a 3:1 ratio, and an offspring of only hermaphrodite individuals. The factor for production of female flowers was designated $S^f$ and the factor for hermaphroditism $S^h$ (Table 1) and it was proposed that $S^f$ is recessive to $S^h$. Furthermore, no male plants resulting from crosses were found. The conversion of male flowers to hermaphrodites by the exogenous application of cytokinin suggested that cytokinins may bypass the genetic regulatory mechanisms of the male determinant as a suppressor of ovule development (*Iizuka & Hashizume, 1968*). *Negi & Olmo (1970)* identified a hermaphroditic male plant, defined by the ability of producing both perfect male flowers and near perfect flowers, capable of fruit production. This hermaphroditic male plant was self-pollinated and the offspring showed a ratio of 3:1 for male and female plants (*Negi & Olmo, 1971*). They selected, at random, some F1 plants and crossed them with a homozygous female *Vitis*. The results showed that, while some crosses resulted in male and female plants, others resulted in only males plants in the F1, meaning

that some males would be homozygous for the male trait (*Negi & Olmo, 1971*). These results also suggested that *vinifera* sex may be inherited as a simple Mendelian factor and, based on this assumption, a new model for sex inheritance was proposed, explained by a single genetic factor with two allelic forms $Su^F$ and $Su^m$ (*Negi & Olmo, 1971*). The allele $Su^F$ represses the development of functional female organs and the allele $Su^m$, in homozygous condition, suppresses the development of male organs. The allelle $Su^F$ is completely dominant over its allele $Su^m$ and the genetic formulae would be $Su^F/Su^m$ or $Su^F/Su^F$ for male and $Su^m/Su^m$ for female (*Negi & Olmo, 1971*) (Table 1). Later, a model that involves a single genetic determinant for sex with three different alleles, *M* for male, *H* for hermaphrodite and *F* for female individuals, with a proposed *M > H > F* dominance pattern (*Antcliff, 1980*) was proposed.

In a study in which several hermaphrodite varieties of *vinifera* were self-pollinated some interesting observations were found (*Bronner, 1981*): (1) some hermaphrodite varieties were heterozygous for the sex locus (resulting 3:1 ratio of hermaphrodites and females), (2) one variety was homozygous (resulting in only hermaphrodite plants), and (3) the variety Muscat Ottonel when self-pollinated resulted in a F1 with a surprising high number of male individuals (*Bronner, 1981*). These observations showed in the two closed *loci* model occasionally crossing-over might occur (depending on the variety in study) or that a more complex model might be needed to explain *Vitis* flower type segregation. In 1983, the previous models from *Valleau (1916)*, *Oberle (1938)* and *Antcliff (1980)* were criticised because they did not not completely explain the total absence asexual individuals (which could be assumed as *So sp/so sp* and *So sp/So sp*) and did not explain the occasional appearance of male plants observed in crosses between hermaphrodites (*Carbonneau, 1983*). Then a more complex model was proposed adding an epistatic gene *E/e* with effects on the alleles H or F, to explain this unpredictable male offspring (*Carbonneau, 1983*). However, is worth to notice that male offspring between hermaphrodites plants were only observe in two different crosses involving four different varieties of which the result offspring from self-pollination is unknown. Therefore, this model has not been tested so far. Instead, the segregation patterns observed in the majority of studies of sex inheritance and recent studies on sex determinism markers appear to support Valleu, Oberle and Antcliff models (*Antcliff, 1980*; *Avramov et al., 1967*; *Dalbó et al., 2000*; *Fechter et al., 2012*; *Hedrick & Anthony, 1915*; *Marguerit et al., 2009*; *Oberle, 1938*).

Wild vines of *V. rotundifolia* are dioecious, while cultivated varieties are hermaphrodite or female, similar to the flower types of cultivated *vinifera* and wild *sylvestris*. There are two sources of hermaphroditism in *muscadinia*, named H1 and H2. The H1 hermaphrodite plant was achieved through the cross of the *muscadinia* female variety Eden and the male variety Mission Male while the H2 hermaphrodite was the result of the cross between the female Scuppernog and the male New Smyrna. These two hermaphrodite plants are the origin of hermaphrodite *V. rotundifolia* (*Loomis & Williams, 1957*). The H1 when self-pollinated originates hermaphrodite and female plants in a 3:1 ratio, which mean that the H1 must be heterozygous HF or *so Sp/so sp*, according to *Antcliff (1980)* or *Oberle (1938)* respectively. However the H2 plant when self-pollinated originates hermaphrodite, female and male plants in a 9:3:4 ratio (*Conner et al., 2017*;

*Loomis & Williams, 1957*). Studies of the records which originate both H1 and H2, lead to the conclusion that they were genetically similar and therefore, H2 could be a mutation of a staminate plant (*Loomis & Williams, 1957*). In *vinifera* the source of hermaphroditism is largely unknown, although the existence of hermaphroditic males (*Negi & Olmo, 1970*, *1971*) and the hermaphrodite offspring between male and female *V. rotundifolia* raises the possibility that a particularly unknown type of *sylvestris* could be in the origins of a full hermaphrodite *sylvestris* and thus target for domestication.

## COMPROMISED POLLEN FORMATION

RNA-Seq studies on *vinifera* and *sylvestris* (male and female) uncovered some genes, whose functionally characterised homologues were previously described to be involved in flower development and organ development, displaying differential expression between male, female and hermaphrodite flower development (*Ramos et al., 2014*, *2017*). These genes belong to several functional categories and seem to be distributed throughout the 19 grapevine chromosomes. Genes responsible for pollen formation seem to have different expression patterns between female and male plants: *LESS ADHERENT POLLEN* (*LAP3*); a gene annotated as a *Nodulin MtN3*, which is a member of the *Lipoxygenase* (*LO*) gene family; a calose synthase 5 (*CalS5*) coding gene; *CYP704B1*; and *ACYL-CoA synthase 5* (*ACOS5*), all have higher expression levels at late development stages of female plants when compared with male or hermaphrodite plants (*Ramos et al., 2014*, *2017*). *Arabidopsis thaliana* mutants for *LAP3* show induced pollen sterility or abnormal exine patterns (*Dobritsa et al., 2009a*). Also, in *A. thaliana*, *MtN3* has a role in exine pattern and in the cellular integrity of microspores (*Guan et al., 2008*). *AtCYP704B1* seems to be implied in sporopollenin synthesis (*Dobritsa et al., 2009b*). Furthermore, mutant plants for *ACOS5* do not produce pollen in late stage of stamens development, as the pollen wall lacks sporopollenin or exine and its formation is thus compromised (*Dong et al., 2005*). *AtCalS5*, which codes for a protein responsible for depositing callose at microspores, is essential for correct exine formation in the pollen, and knockout mutants for *CalS5* resulted in reduced fertility (*Dong et al., 2005*; *Enns et al., 2005*). However, as a whole, the above experiments result in either the absence or defective expression of these genes. In female *sylvestris* the opposite was observed, with the transcripts being highly expressed at late flower development. The effect of over-expression of these genes has not yet been studied, nevertheless, the results might suggest that excess of sporopollennin and exine deposition could result in infertile pollen. A large deposition of these pollen wall elements might lead to small and oval pollen grains lacking pores (*Gallardo et al., 2009*). In *sylvestris* female pollen is also smaller and fewer grains are formed (*Gallardo et al., 2009*). This might be due to resource allocation in pollen development, since excessive deposition of sporopollenin and exine might consume more resources, leading to the formation of fewer and smaller pollen grains in female plants. Also, the *VviERECTA* gene seems to have less expression, at late development stages, in female when compared with male and hermaphrodite flowers. In *Arabidopsis*, *ERECTA* mutants are associated with short thick stamens and pedicels (*Ferrandiz, Pelaz & Yanofsky, 1999*; *Torii et al., 1996*). This phenotype is not observed in female *Vitis* plants, although their stamens are shorter

than those of male and hermaphrodite plants. Also, in situ hybridisation targeting *VviERECTA* mRNA showed that this gene is expressed similarly in all three flower types in the stamens and carpels (*Coito, 2018*).

Pollen wall appears to be compromised by the over-expression of transcripts involved in its formation. However, transcription factors might also play a role in female stamens phenotype *TRANSCRIPTION FACTOR ABORTED MICROSPORES-LIKE* (*AMS*) (*Ramos et al., 2017*), which only shows expression in female plants at late developmental stages and thus could contribute to pollen inviability. VIT_212s0057g00440 (*Ramos et al., 2017*), which is over-expressed in flower plants at stage H, is annotated as an "acyl-n-acyltransferase domain-containing protein" and may also be contributing to the abnormal pollen wall formation and subsequent infertility.

## THE DEVELOPMENT OF THE MALE CARPEL: POSSIBLE HORMONE OR TIMEFRAME CONTROL

The RNA-Seq studies (*Ramos et al., 2014*, *2017*) also showed a small number of genes related to carpel formation differently expressed in late stages of male flower development. While *VviJAGGED* and *VviERECTA* were highly expressed at late development stages in male plants, *VviSHEPHERD* showed the opposite pattern, being less expressed at these stages in male plants when compared to the other flower types. Other gene involved in carpel formation, *VviCRABSCLAW*, seems to be less expressed in late development stages in male flowers. In *Arabidopsis JAGGED* codes a putative $C_2H_2$ zinc-finger transcription factor (*Ohno et al., 2004*) and in stamens, *AtJAGGED* expression promotes the growth of the adaxial side, a requisite for microsporangia formation, while in the carpel *AtJAGGED* acts by promoting the growth of the valves that enclose the ovules (*Dinneny, Weigel & Yanofsky, 2006*). *AtERECTA* codes for a leucine-rich receptor-like serine/threonine kinase (*Torii et al., 1996*) and its function includes specifying aerial organ shape and size, mainly by promoting cell proliferation (*Van Zanten et al., 2009*). This is evident in *AtERECTA* mutants, which show a dwarfish morphology (*Shpak, Lakeman & Torii, 2003*, *Shpak et al., 2004*). However, as stated previously, in situ hybridisation targeting *VviERECTA* mRNA showed no differences between flower types (*Coito, 2018*).

In *Arabidopsis*, *SHEPHERD* seems to be required for the correct fold of CLAVATA proteins, which in turn promotes the differentiation of peripheral stem cells during organ initiation (*Ishiguro et al., 2002*). However, in *Vitis*, several *CLAVATA* genes seem to be differently expressed during development but not between flower types. *CRABSCLAW* is a member of the YABBY family (*Bowman & Smyth, 1999*; *Siegfried et al., 1999*) and seems to be required for nectaries development independently of *AGAMOUS* activity (*Baum, Eshed & Bowman, 2001*). In *Arabidopsis*, *CRABSCLAW* mutations affect the gynoecium growth resulting in a shorter carpel (*Alvarez & Smyth, 1999*). In grapevine male plants seem to have a shorter carpel, however, *VviCRABSCLAW* is expressed identically in the nectaries and carpel in all three flower types (*Coito, 2018*). This observation shows that differences in carpel formation between male and female plants of *sylvestris* occur very late in flower organ development.

A search for transcripts involved in phytohormones synthesis and signalling found several differentially expressed transcripts between flower types during development. The expression of genes involved in hormone biosyntesis and signalling, as well as the expression of *VviJAGGED*, *VviERECTA* and *VviSHEPHERD*, seems to be more associated with the timing of flower development initiation rather than with the development of specific flower organs. However, some hormone related genes could be involved in carpel shape regulation in male flowers, since they are differentially expressed in late stages of flower development. *VviPINOID* and *VviETTIN* were both more expressed in male plants at late flower development stages. In *Arabidopsis*, *PINOID* codes a serione-threonine kinase and its over-expression is linked to negative regulation of auxin signalling, which results in a dwarf plant morphology (*Christensen et al., 2000*) and *AtETTIN*, which codes for a protein with a DNA-biding domain highly similar of the *AUXIN RESPONSIVE FACTOR 1* involved in early response to auxin (*Ulmasov, Hagen & Guilfoyle, 1997*, *Ulmasov et al., 1997*). *ETTIN* defective *Arabidopsis* mutants show enhanced proliferation of style and internodes (*Nemhauser, Zambryski & Roe, 1998*). The effects of the over-expression of *ETTIN* are largely unknown; however, it could lead to a diminished proliferation of style or, coupled with higher expression of *PINOID*, could lead to a deficient response to auxin signalling, in turn leading to an abnormal carpel with underdeveloped style and stigma. Thus, these genes could act together through an auxin regulated process in carpel development (*Ferrandiz, Pelaz & Yanofsky, 1999*) and it is quite likely that hormones influence carpel morphology. As previously demonstrated, the exogenous application of the synthetic cytokinin BAP (6-benzylamino-9-(2-tetrahydropyranyl)-purine), induces sex reversion of *sylvestris* male flowers into hermaphrodites (*Negi & Olmo, 1971*). Similarly, more recently, the exogenous application of a cytokinin in *Sapium sebiferum* induced a female flower development, instead of a male one (*Ni et al., 2018*). In *A. thaliana*, *AtAPRT1* mutants develop male sterility due to the formation of atypical pollen (*Gaillard et al., 1998*; *Moffatt & Somerville, 1988*). Taking into consideration the close cross talk between auxins and cytokinins, it is not to exclude that flower type and sex specification may be controlled through hormone regulation. However, it is difficult to distinguish normal floral development pathways from the abnormal carpel formation through this approach, since these pathways seem dependent on an expression balance of hormone related genes.

## HOMEOTIC GENES IN GRAPEVINE

The ABCDE model postulates that genes from three regulatory functions act in concert to confer organ identify during flower development, subdividing the developing flower into whorls (*Coen & Meyerowitz, 1991*). For example, in *A. thaliana*, *APETALA1* and *APETALA2*, both A-function genes, specify sepal identity, while A-function genes combined with B-function genes (*APETALA3* and *PISTILLATA*) confer petal identity. B-function genes together with C-function genes (*AGAMOUS*) confer stamen identify and *AGAMOUS* alone confers carpel identity (*Coen & Meyerowitz, 1991*). The D-function (composed by the genes *SEEDSTICK* and *SHATTERPROOF 1* and *2*) is required for ovule identity (*Pinyopich et al., 2003*) and E-function genes (conferred by four *SEPALLATA*

genes) are redundantly necessary for the specification of the four flower whorls (*Ditta et al., 2004*; *Favaro et al., 2003*; *Pelaz et al., 2000*).

In hermaphrodite grapevine, in situ hybridisation (*Coito et al., 2018*) and real-time PCR experiments (*Palumbo et al., 2019*) showed that B-class genes are expressed in petals, stamens and carpel in the early stages of development but in later stages, *VviPI* seems to stop being expressed in the forth whorl while *VviAP3* and *VviTM6* still display some expression in the carpel. In the case of wild grapevine a possible abnormal expression of ABCDE model genes, particularly those responsible for stamens and carpel organ identity could trigger the abnormal morphologies that distinguish female and male flowers.

Upon the emergence of sepal primordia, *VviPI* and *VviAP3* are excluded from the sepals. Afterward, these genes are expressed in the third whorl and continue throughout flower development (*Coito et al., 2018*). Other important gene to refer is *VviSUPERMAN* (*VviSUP*) which was found to clearly mark the boundary between the third and fourth whorl, equally in both female and male flowers (*Coito et al., 2018*). Finally, the C-function gene, *VviAG* seems to be expressed at the centre of the flower meristem and during flower development it is gradually excluded from the first and second whorl, while retaining its expression in the stamens and carpel in both female and male plants (*Coito et al., 2018*).

This expression profiles excludes the possibility that B- and C-class genes may be responsible for carpel or stamen abortion and, therefore, controlling flower type specification.

## A SMALL GENOMIC REGION DEFINING THE SEX IN GRAPEVINE

Several attempts were made to understand and provide insight into the molecular mechanism regarding the origins of sexual dimorphism present in *sylvestris* individuals. Several genetic mapping studies based on the 8× version of the *Vitis* genome annotation (http://www.genoscope.cns.fr/externe/GenomeBrowser/Vitis/entry_ggb.html) (*Dalbó et al., 2000*; *Marguerit et al., 2009*; *Riaz et al., 2006*) located the *locus* responsible for sex determination at the vicinity of the genetic markers VviMD34 and VviIB23 on chromosome 2 in the 8X version (http://genomes.cribi.unipd.it/grape/). With the use of these genetic markers and genetic maps, the authors conclude that only one gene would be responsible for the dimorphism in *Vitis* (*Dalbó et al., 2000*; *Marguerit et al., 2009*; *Riaz et al., 2006*). This conclusion was in accordance with the work of *Antcliff (1980)* in which *Vitis* flower type trait follows a simple Mendelian inheritance model whit three alleles: male (M), female (F) and hermaphrodite (H).

Using these markers putatively linked to the sex *locus*, a new genetic map was developed and refined to restrict the sex *locus* to 143 kb in the chromosome 2, between 4,907,434 and 5,050,616 bp (*Fechter et al., 2012*). This region contained 16 genes, including an *ADENINE PHOSPHORIBOSYILTRANSFERASE* (*VviAPRT*) which was identified as a possible genetic marker able to discriminate female plants from male and hermaphrodite ones (*Fechter et al., 2012*). The suggestion that *VviAPRT* could potentially be involved in sexual dimorphism in *Vitis* is particularly interesting, since in other species *APRT* homologues code for a key metabolic enzymes participating in cytokinin metabolism

(*Allen et al., 2002*; *Mok & Mok, 2001*). A more recent study, focusing on the 143 kb region of chromosome 2 extended the sex *locus* region to 158 kb downstream of the genetic marker VviIB23 and encompassing the previous 143 kb region (*Picq et al., 2014*). This new *locus* showed haplotype diversity, linkage disequilibrium and several genes segregating to typically associated X-Y sex determining region, with males being XY and females XX with one of them being the previously identified *VviAPRT* (*Picq et al., 2014*). Also, it has been suggested that many of the genes in the sex *locus* could have homologue sequences located in the grapevine virtual chromosome unknown (*Picq et al., 2014*), thus potentially being in a heterozygous state in *vinifera* reference genome, suggesting that the reference genome PN40024 could be heterozygous regarding the sex *locus* (*Jaillon et al., 2007*; *Picq et al., 2014*). Similar results were found for *V. rotundifolia* (*Lewter et al., 2019*). Linkage maps were constructed for two F1 populations segregating for flower type and berry colour. Results showed that the putative sex *locus* on *V. muscadinia* mapped on chromosome 2 of *vinifera*, in the previous identified sex *locus* (*Battilana et al., 2013*; *Dalbó et al., 2000*; *Fechter et al., 2012*; *Marguerit et al., 2009*; *Picq et al., 2014*; *Riaz et al., 2006*). This suggests that the sex *locus* might be conserved among *Vitis* species.

In 2017, *Zhou et al. (2017)*, using NGS (Next Generation Sequence), identified two regions on the chromosome 2 that might be linked to sex determination in grapevine. One region contained six genes expressed in female *sylvestris*, while the second region is believed to harbour a dominant female sterility gene(s) (*Zhou et al., 2017*). These results seem to suggest that two factors might be involved in flower type determination in grapevines. A study which though to identify structural variances in *vinifera* (*Zhou et al., 2019*) shed some light into *V. vinifera* sex determination and domestication. Important variations were found between cultivated and wild grapevines in the potential sex *locus*. A phylogeny build on the sex determinant region showed that the hermaphrodite haplotype was closer to the *M* haplotype than to the female. It also showed cultivated varieties homozygous for the hermaphrodite trait, and varieties heterozygous, having the hermaphrodite and female structural variant (*Zhou et al., 2019*). This corroborates the early Mendelian studies in grapevine which states that some cultivated grapevine varieties could either be *FH* or *HH* (*Antcliff, 1980*; *Avramov et al., 1967*; *Oberle, 1938*; *Valleau, 1916*). This study also shows that wild grapevine can also be *HM* as proposed by *Valleau (1916)* and that female grapevines are homozygous for the female trait (*Antcliff, 1980*; *Avramov et al., 1967*; *Oberle, 1938*; *Valleau, 1916*; *Zhou et al., 2019*).

Recently, two genetic markers were found, one marker in the *VviAPRT3* gene and a new marker called *VviFEMALE SEX (VviFSEX)* gene (*Coito et al., 2017*). *VviFSEX* gene is located in the previously identified putative sex *locus* in the grapevine chromosome 2 (*Fechter et al., 2012*; *Picq et al., 2014*), downstream of the *VviAPRT3*. *VviFSEX* is also one of the six genes expressed in one of the regions linked to sex determination in grapevine (*Zhou et al., 2017*). When used together, the amplification profile of these two genetic markers made possible to distinguish female and male *sylvestris* and hermaphrodite *vinifera*. It was also showed that male plants where heterozygous for both markers, while female plants were homozygous. Hermaphrodite plants were homozygous for the *VviAPRT3* genetic marker but heterozygous for the *VviFSEX* marker (Fig. 5). In addition,

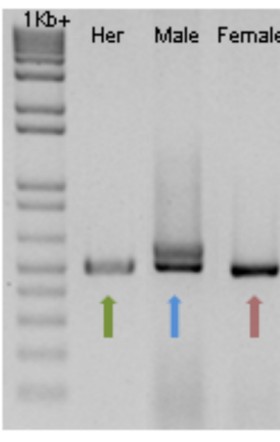

**Figure 5 Schematic of the amplification of *VviFSEX* and *VviAPRT3*.** Hermaphrodite (Her) plants show only one amplification fragment of *VviAPRT3* but two fragments of *VviFSEX*, male plants show two fragments of both *VviAPRT3* and *VviFSEX*, while female plants only show one amplification fragment of each genetic marker, adapted from *Coito et al. (2017)*. According to *Oberle (1938)*, and through this amplification profile, Her plants would be homozygous recessive for *SO* (*so so*) and heterozygous for *SP* (*Sp sp*), or HF, according to *Antcliff (1980)*, likewise male plants would be heterozygous for both *SO* (*So so*) and *SP* (*SP sp*) or MF. Finally, female plants would be homozygous recessive for both *SO* (*so so*) and *SP* (*sp sp*), or FF.

using markers developed from *Picq et al. (2014)*, (VSVV006 and VSVV009) (*Conner et al., 2017*) in two F1 populations obtained by crossing three *V. rotundifolia* varieties segregating for flower type and berry colour, it was possible to identify female, male and hermaphrodite plants (*Lewter et al., 2019*).

## THE MOLECULAR VALIDATION OF THE LONG-SOUGHT SEX *LOCUS*

Two seemingly contrasting hypothesis exist, as previously referred: (1) two linked genes (*Oberle, 1938*) and (2) one gene, three alleles (*Antcliff, 1980*). With the second hypothesis being somewhat more accepted in recent works (*Dalbó et al., 2000*; *Fechter et al., 2012*; *Picq et al., 2014*) while the first hypothesis is suggested based on the evolutionary assumption that two mutations are required in a hermaphrodite ancestral population in order for male and female individuals to emerge (*Charlesworth & Charlesworth, 1978*) and seems to find some validation on recent works (*Coito et al., 2017*; *Ramos et al., 2014*; *Zhou et al., 2017*). With the development of the two genetic markers (*Coito et al., 2017*) it became possible to associate these to the previous observations by *Oberle (1938)*, *Valleau (1916)*, *Avramov et al. (1967)* and *Antcliff (1980)*, therefore validating them by a molecular approach.

The combinations of the two genetic markers could represent or hint the *locus* responsible for male and female plants and contributed to the possibility that grapevine sex *locus* does reside in the chromosome 2. In the two linked genes hypothesis (*Oberle, 1938*) the combination *SO/SP* identify a male plant, while the allele combination *so/SP* identify a hermaphrodite plant and the combination *so/sp* identify a female plant. In the recent finding, different combinations of the molecular markers *VviAPRT3* and *VviFSEX* can also identify male, female and hermaphrodite plants (Fig. 5), however is unlikely that these markers are solely responsible for sex specification on grapevines, rather, they seem to identify a large *locus*, or genomic region responsible for such dimorphism. Taking this into consideration the combination of both markers can also be applied to the one gene, three alleles hypothesis (*Antcliff, 1980*), where the molecular markers combinations could represent and identify the M, H and F *locus* (Fig. 5).

## *VITIS* FLOWER TYPES: A LONG ROAD TO GO

Despite the most recent molecular studies corroborate the Mendelian works on grapevine flower type segregation, the knowledge is still limited. Likewise, there are many gaps in understanding *Vitis* flower types development and regulation. Genetic markers (*Battilana et al., 2013*; *Dalbó et al., 2000*; *Fechter et al., 2012*; *Marguerit et al., 2009*; *Riaz et al., 2006*) point to a precise region of chromosome 2 which encompass several genes between 4.09 and 5.04 Mbp. However, despite the development of genetic markers that can differentiate hermaphrodite, male and female plants (*Coito et al., 2017*; *Conner et al., 2017*; *Picq et al., 2014*), sequence analysis (*Fechter et al., 2012*; *Picq et al., 2014*; *Ramos et al., 2014*, *2017*) has been unable to identify the key players responsible for the regulation of flower types. Therefore it is still unknown which gene or genes, trigger exactly the formation of the different types of flowers. Renewed efforts in sequencing *sylvestris* genome and fine mapping the sex region could shed light regarding the differences between putative key players regulating flower development in male and female.

The use of genetic editing technology can also be of assistance in the comprehension of grapevine flower type determination. CRISPR/Cas9 technology was already tested with success in tomato (*Solanum lycopersicum*) targeting the gene *SlAGO7*, whose mutant can be easy identified. Loss of function mutants for *SlAGO7* show leaflets without petioles and later leaves lacking laminae (*Brooks et al., 2014*). RNA interference (RNAi) is another technology able to explore the putative function and role of candidate genes in grapevine flower type specification. RNA interference has been previously used to down-regulate *OsSQS* which resulted in a reduced stomatal conductance and therefore higher drought resistance (*Manavalan et al., 2012*). It was used to shut down the gene *Aminocyclopropane-1-carboxylate* in tomato plants and these transgenic plants produced fruits capable of releasing traces of ethylene and so had a longer shelf life (*Xiong et al., 2005*).

Naturally occurring mutations for flowering in grapevines, conversely to *fleshless* berry mutant applied to identify genes putatively involved in the early development of grapevine fruit (*Fernandez et al., 2007*), are still poorly documented. Although natural mutations are important tools to provide information on *Vitis* flower segregation, few flower mutations have been reported. Flower mutants with abnormal petals and stamens were

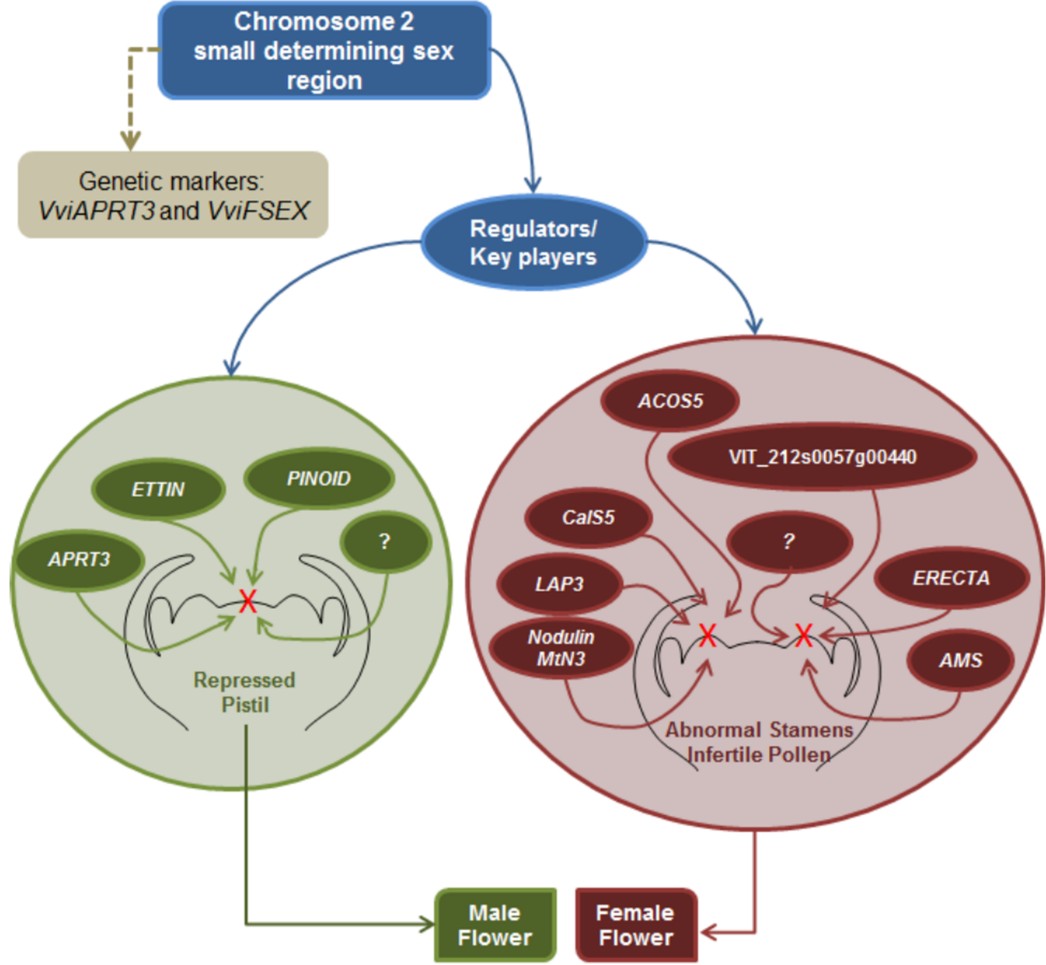

**Figure 6 Scheme of the putative mechanism regulating dimorphic flower development in *V. v. sylvestris*.** A small determining sex region exists in the chromosome 2, in which *VviAPRT3* and *VviF-SEX* are genetic markers defining male or female individuals. This small sex determining region has one or several key regulator genes which will activate a cascade of pathways that take place downstream of the onset of *VviPI*, *VviAP3* and *VviAG* to form either male or female plants. Depending of the key players in the sex defining region, *ETTIN*, *PINNOID*, *APRT3* and other unknown players contribute to the abnormal development of the pistil in male plants, Likewise, *LAP3*, *CalS5*, *AMS*, *ERECTA*, *ACOS5*, *Nodulin/MtN3*, VIT_2012s0057g00440 and possible other key players induce excess of exine and sporopollenin in the pollen wall and so becomes infertile, in female plants.

first reported by *Sreekantan et al. (2006)*. One mutant showed normal sepals and petals, but the carpel structures appeared fused to pseudo-ovules in the place of stamens and lacked ovary in the central whorl. The flowers of this mutant were not functional. The second flower mutant described (*Sreekantan et al., 2006*), showed normal stamens number but filaments were shorter or absent with only 2–5% of flower developing into seeded berries. This second flower mutant develops in a similar way of female flowers; however, only *VviPI* was characterised in these mutants (*Sreekantan et al., 2006*) and no F1 segregation was studied. More six grapevine flower mutants were described and characterised (*Chatelet et al., 2007*), although the expression of only some homeotic genes

was covered in the study. Additionally, two grapevine rootstocks could only be identified through flower morphology (*Wolf et al., 2003*), variety SO4 which shows only male flowers, and the variety Binova that produces hermaphrodite flowers. AFLP and SAMPL markers failed to distinguished these two rootstocks, leading the authors to the conclusion that Binova could be a somatic mutant of SO4 (*Wolf et al., 2003*). Unfortunately, no self-pollination of Binova was reported so far, and so, it is unknown how a F1 segregates for flower type.

## CONCLUSIONS

Sex specification in *Vitis* continues to be an elusive subject; however, this review seeks to gather all the information regarding sex specification and early identification in *vinifera*, from the first Mendelian genetic studies to more recent molecular and NGS approaches. It seems that there is a consensus regarding male determinant being dominant over hermaphrodite and female plants, while hermaphrodite determinant seems to be dominant over the female determinant. Mendelian genetic studies regarding grapevine offspring showed that female plants were homozygous while hermaphrodite and male grapevine could either be homozygous or heterozygous. Several studies regarding molecular and NGS approaches resulted in the finding of a potential sex *locus* located in the chromosome 2 and two genetic markers being developed to identify male, female and hermaphrodite plants and validate the previous assumptions regarding the homozygosity or heterozygosity of male, female and hermaphrodite plants. Nevertheless, is still unknown if sex specification is due to one or more genes, only that is most likely present in the chromosome 2 sex *locus* and that it seem to influence several genes downstream of the ABCDE homeotic genes mainly pollen development genes and hormone signalling genes (Fig. 6).

### Funding

This work was supported by the funded project PTDC/AGR-GPL/119298/2010 from Fundação para a Ciência e Tecnologia (FCT, Portugal), by UID/AGR/04129/2013 centre grant from FCT, Portugal (to LEAF) and by PEst-OE/BIA/UI4046/2014; UID/MULTI/04046/2013 centre grant from FCT, Portugal (to BioISI) and are supported by FCT fellowships JL Coito, MJN Ramos, H Silva, M Rocheta, respectively, SFRH/BD/85824/2012, SFRH/BD/110274/2015, SFRH/BD/111529/2015, SFRH/BPD/64905/2009. The funders had no role in study design, data collection and analysis, decision to publish, or preparation of the manuscript.

### Grant Disclosures

The following grant information was disclosed by the authors:
Fundação para a Ciência e Tecnologia (FCT, Portugal): PTDC/AGR-GPL/119298/2010.
FCT, Portugal (to LEAF): UID/AGR/04129/2013.
FCT, Portugal (to BioISI): PEst-OE/BIA/UI4046/2014; UID/MULTI/04046/2013.

FCT: SFRH/BD/85824/2012, SFRH/BD/110274/2015, SFRH/BD/111529/2015, SFRH/BPD/64905/2009.

## Competing Interests

The authors declare that they have no competing interests.

## Author Contributions

- João L. Coito analyzed the data, prepared figures and/or tables, authored or reviewed drafts of the paper, approved the final draft, searched the database.
- Helena G. Silva analyzed the data, prepared figures and/or tables, authored or reviewed drafts of the paper, approved the final draft, searched the database.
- Miguel J.N. Ramos analyzed the data, authored or reviewed drafts of the paper, approved the final draft, searched the database.
- Jorge Cunha analyzed the data, contributed reagents/materials/analysis tools, authored or reviewed drafts of the paper, approved the final draft, searched the database.
- José Eiras-Dias analyzed the data, contributed reagents/materials/analysis tools, authored or reviewed drafts of the paper, approved the final draft, searched the database.
- Sara Amâncio analyzed the data, authored or reviewed drafts of the paper, approved the final draft, searched the database.
- Maria M.R. Costa analyzed the data, authored or reviewed drafts of the paper, approved the final draft, searched the database.
- Margarida Rocheta analyzed the data, prepared figures and/or tables, authored or reviewed drafts of the paper, approved the final draft, searched the database.

## Data Availability

There is no raw data; this is a literature review.

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
