# Peer review of "Vitis flower types: from the wild to crop plants"

_PeerJ, doi:10.7717/peerj.7879_

## Round 0.1 · original submission · Major Revisions

Your manuscript has been seen by two reviewers. Based on their detailed assessment and my own, I feel this manuscript could be suitable for publication pending a number of major revisions. In particular, the authors should expand their review to address the four major criticisms outlined by the second reviewer. Reviewer 2 points out that there are still gaps in our understanding of grapevine flower types, but this review does a nice job summarizing what is known.

Reviewer 1 ·

Basic reporting

In this manuscript (peerj-#37587), Coito and colleagues reviewed the genetics of sex determination in Vitis with male, female and hermaphrodite flowers. The authors started from famous historical phenotypic studies which established the basic theories of grapevine sex determination. Recent QTL, RNA-seq and NGS analyses were compared and integrated with early scenarios. In general, this is a careful deep review that is worth sharing with the grapevine community and the sex evolution community, however, I also have the following comments for the authors to consider in revision.

Experimental design

The literature reviewed here is complete with no bias and the historical and current investigations were incorporated very well. The writing is clear too.

Validity of the findings

The conclusions are well justified based on old and new studies.

Additional comments

Major:

1) Please write the species name correctly. Both vinifera and sylvestris are subspecies of Vitis vinifera. It should be written as Vitis vinifera subsp. vinifera and Vitis vinifera subsp. sylvestris, respectively. The author could use abbreviation after the first reference, for example, V. vinifera subsp. sylvestris (hereafter sylvestris).

2) There is very recent study using genomic sequences across tens of wild and cultivated grapevines to study the sex determination region (doi: https://doi.org/10.1101/508119), which should be included in this review because it explains some of the old theories on grapevine sex very well. The paper found important structural variations in the sex region and the phylogeny showed very interesting patterns.

Minor:

L49-53: The authors might want to update this information from current literature. The newest fossil record suggested that grapevine domesticated from their wild ancestor, the wild Eurasian grapevine (sylvestris), nearly ~8,000 years ago in the Transcaucasus (https://doi.org/10.1073/pnas.1714728114). And the only evolutionary genomic analyses suggested the human managed of wild grapevine population 22000-30000 years ago (https://doi.org/10.1073/pnas.1709257114). This is reasonable because humans should have eaten fruits before getting drunk.

L101: “and a only” should be “and an only”

L108: “XX century”: what is the XX?

L124: “has been” should be “have been”?

L134-145: The authors may want to compare these pattern with the phylogeny I mentioned above.

L326-328: This sentence is hard to follow. Please rephrase it.

L378-380: Please check whether the structural variation is in line with this observation (doi: https://doi.org/10.1101/508119).

L394: “sex locus do” should be “sex loci do” or “sex locus does”

L396: “an hermaphrodite plant” should be “a hermaphrodite plant”

L400: “indentify” should be “identify”

L408: “mendelian” should be “Mendelian”

L419: “hormone signalling” should be “hormone signaling”

Figure 2 legend: “Stout (1921 )” should be “Stout (1921)”

Figure 4 legend: “(1938 )” should be “(1938)”

Figure 5 legend: (Coito et al. 2017). According to (Oberle 1938)

Figure 5 legend: (Antcliff 1980)

Figure 5 legend: “heterozygous for both SO (So so) and SP (SP SP)” should be “heterozygous for both SO (So so) and SP (Sp sp)”

Table 1 legend: “hermaphrodites, males and females plants” should be “hermaphrodite, male and female plants”

Reviewer 2 ·

Basic reporting

The abstract contends that the manuscript will address the hermaphrodite V. vinifera sylvestris as a wild plant or as a progenitor to domesticated grapevine, but this is barely mentioned in the body of the manuscript and certainly not as a testable hypothesis (lines 30-31).

When referring to a botanical family, the -aceae ending on the family name indicates "family", so please use "Vitaceae" and not "Vitaceae family" (lines 41, 45).

Line 42: Northern Hemisphere

At the beginning of a sentence, always spell out the genus name (line 43), such as "Vitis vinifera . . . "

Line 44: There may be only one Vitis species native to Europe, but there are many species (possibly more than 40) native to Eurasia, as Eurasia encompasses all of Asia as well as all of Europe. Change line 44 to read ". . . in Europe".

Line 62: “14 genera” not “14 genus”.

In referring to the chromosome number, use 2n = 2x = 38 for bunch grapes (subgenus Vitis) and 2n = 2x = 40 for muscadines (subgenus Muscadinia).

The discussion of the Vitaceae should address the current and recent work on Muscadinia flower sex inheritance and mapping:
Lewter, Jennifer, Margaret L. Worthington, John R. Clark, Aruna V. Varanasi, Lacy Nelson, Christopher L. Owens, Patrick Conner, and Gunawati Gunawan. "High-density linkage maps and loci for berry color and flower sex in muscadine grape (Vitis rotundifolia)." Theoretical and Applied Genetics (2019): 1-15.

Crosses between V. vinifera and V. rotundifolia (line 66) typically are viable, in the sense that the seeds germinate and the plants grow and develop), but they are of reduced viability in the sense that the first generation hybrid vines are usually of very low fertility, with little or no set of viable seeds. It’s inaccurate to flatly state that the hybrids are viable when this is not true; substantial qualification is needed to describe the deficiencies in the first generation hybrids. The cross between V. vinifera and V. rotundifolia is harder to successfully conduct compared to intraspecific crosses. The seed set is lower when crosses between these species are performed than intraspecific crosses.

The text on the polyploid origin theory (lines 69 – 80) of Muscadinia and Vitis is interesting and accurate, but it is basically not relevant to the theme of the manuscript.

Line 197:
Carbonneau (1983) may report occasional staminate flowered seedlings in crosses of perfect flowered vines. However, Bronner (1981) reported finding 11 staminate flowered (male) vines in a self-pollinated population (total population size = 37) from Muscat Ottonel (Bronner, A. "Observation de types sexuels males dans la descendance par autofecondation de la variete Muscat Ottonel (Vitis vinifera L.)." Vitis (1981)). Since so much of this manuscript is given to distinguishing among the models for flower type inheritance (two tightly linked loci vs one locus with three alleles), Bronner’s findings must be addressed, because these surely inform the discussion about the development and validation of the flower type models. My reading of Bronner is that these findings indicate that there are two tightly linked loci, but in the case of Muscat Ottonel there is occasionally cross over, leading to the observation of staminate flowered vines.

Line 190:
However, the homozygous male plants are found, in direct contradiction to what is reported in the manuscript--this is found by Negi and Olmo, 1971. They created homozygous male plants and they crossed them to female plants and they got male progeny only--this is shown in their paper. See Table 4 in Negi and Olmo, 1971, page 270. You might not believe what is reported, but they do report that there were male flowered vines that when crossed to female vines gave only male flowered seedlings.

Experimental design

No comment.

Validity of the findings

These key conclusions or findings listed below are missing and they are major deficiencies in the state of the knowledge of the molecular genetics of grapevine flower type control:
1) There is no sequence difference—whether SNP or structural variant, insertion, deletion, or transposition—that is shown to be causally related to flower sex differences across genotypes. Linked markers and candidate genes are interesting, but this does not provide validation for or against a particular model and certainly does not evaluate the candidate genes.
2) There is no gene editing, recombinant DNA, or mutagenesis validation experiment reported here to validate the candidate genes for flower type. Did anyone use RNAi (or any other method) to turn on or off candidate genes to determine the impact on phenotype, for example? Did anyone apply mutagenesis and then report flower type mutants in a population?
3) The discussion of flower type mutants is totally absent. Binova is a perfect flowered mutant of SO4, while SO4 is ordinarily male flowered. Did anyone check the sequence of Binova to compare it with SO4? Did anyone observe segregation of flower type in Binova progeny? There are two rootstocks (Star 50 and Star 74) which are Binova seedlings from self-pollination, but what is the segregation of flower type in Binova self-pollinated populations (if any). The Binova should be self-pollinated and cross pollinated, but this is unreported as far as I know.
4) The origin of perfect flowered Muscadinia is well documented, but ignored here. For a review of grapevine flower type genetics, this must be addressed. There are two different flower type inheritance modalities in Muscadinia. There are probably two different loci for flower type in subgenus Muscadinia—in contrast to one locus in subgenus Vitis. That’s very, very important for understanding the flower type pathway.

Additional comments

I suggest that the time is not right for a review of grapevine flower type. The manuscripts cited in this review are complete and can be easily read and understood. However, this review does not substantially improve our understanding of the current state of this particular topic.

---

## Round 0.2 · Minor Revisions

Your revised manuscript has satisfied the previous concerns raised by the two reviewers and represents a nice summary of sex determination systems in grape vine. There are few more minor points that should be addressed before this manuscript is accepted.

Reviewer 1 ·

Basic reporting

no comment

Experimental design

no comment

Validity of the findings

no comment

Additional comments

In this revised manuscript the authors have clearly addressed all the concerns raised. I think this is now acceptable for publication and will be of immense importance to the field. I would like to congratulate the authors on a job well done.

Reviewer 2 ·

Basic reporting

No comment.

Experimental design

No comment; this is a review manuscript.

Validity of the findings

No comment.

Additional comments

Thanks to the authors providing more information, the revised manuscript is distinctly improved from the first submission.

When referring to the species Vitis rotundifolia, this may be called V. rotundifolia (italics) or by the common name “muscadine”, but the species “V. muscadinia” is not correct.

The abstract states that three flower types can be observed: hermaphrodite in vinifera and male or female flowers in sylvestris. This is incomplete—there are certainly pistillate flowered vinifera varieties, such as Grk and Picolit (wine grapes) and Hunisa, Chaouch, Nimrang, and Almeria (table grapes), among others. These are completely vinifera, not sylvestris, although it is unusual—and certainly inconvenient from a horticultural perspective. It is more complete to state that vinifera could have hermaphrodite or female flowers and sylvestris could have male or female flowers.

Line 45: Vitaceae, not Vitacea

Line 137: Use “reflexed” in place of “downright”

Lines 141-142: Explain what is meant by FFM. I understand FM, which is a male flowered vine, heterozygous for the pistillate female flower F allele. What is FFM? We should expect two alleles only.

Lines 165-167: Explain why a cross between heterozygous hermaphrodites and homozygous Sp male plants should originate male and hermaphrodites in a 1:1 ratio.
We can assume that a heterozygous hermaphrodite is HF (so SP/so sp). The homozygous male is MM (or SO SP/SO SP). Since M is dominant to H and F, the cross of the homozygous male to any other flower class (hermaphrodite classes HH and HF, female class FF) can only produce vines that carry the dominant M (or SO SP) allele and are male flowered. There should be no segregation observed in the offspring of homozygous males, unless there is recombination. Address this.

Line 182: was self-pollinated

Line 205:
Explain the importance of asexual individuals—are these observed? Does Carbonneau report them? What are the characteristics of asexual individuals? If there are male vines occurring in seedling populations from hermaphrodite crosses (with no male vine parent), are there asexual individuals as well?


Lines 213-214: Suggested text:
Wild vines of V. rotundifolia are dioecious, while cultivated varieties are hermaphrodite or female, similar to the flower types of cultivated vinifera and wild sylvestris. There are two sources of hermaphroditism. . .

Line 397: The observation that chromosome 2 hosts a flower type locus in V. vinifera and in V. rotundifolia does suggest that the flower type locus may be conserved among Vitis species. However, what evidence is there that this flower type locus is conserved among the Vitaceae? For the flower type locus to be conserved among the Vitaceae, I would expect to see documentation of flower type locus on chromosome 2 or its equivalent in another genus (or preferably genera) in the Vitaceae, such as Cissus or Causonis. Better to rewrite the sentence to read:
This suggests that the sex locus might be conserved among Vitis species.

---

## Round 0.3 · accepted · Accept

Your revised manuscript has addressed the remaining comments and concerns from the second reviewer and is now suitable for publication in PeerJ.